# A potential explanation for the global increase in tropical cyclone rapid intensification

Kieran Bhatia [1] ✉, Alexander Baker [2], Wenchang Yang [3], Gabriel Vecchi [3,4], Thomas Knutson [5], Hiroyuki Murakami [5], James Kossin [6], Kevin Hodges [2], Keith Dixon [5], Benjamin Bronselaer [7] & Carolyn Whitlock [8]

Tropical cyclone rapid intensification events often cause destructive hurricane landfalls because they are associated with the strongest storms and forecasts with the highest errors. Multi-decade observational datasets of tropical cyclone behavior have recently enabled documentation of upward trends in tropical cyclone rapid intensification in several basins. However, a robust anthropogenic signal in global intensification trends and the physical drivers of intensification trends have yet to be identified. To address these knowledge gaps, here we compare the observed trends in intensification and tropical cyclone environmental parameters to simulated natural variability in a high-resolution global climate model. In multiple basins and the global dataset, we detect a significant increase in intensification rates with a positive contribution from anthropogenic forcing. Furthermore, thermodynamic environments around tropical cyclones have become more favorable for intensification, and climate models show anthropogenic warming has significantly increased the probability of these changes.

Rapid intensification (RI; defined as the 95th percentile of 24-h intensity changes[1]) can quickly transform a tropical cyclone (TC) from a relatively predictable natural hazard (such as heatwaves) with skillful long-range forecasts to an unpredictable one (such as tornadoes) with reliable warnings only hours in advance. Over the last five years, guidance models and National Hurricane Center (NHC) forecasts have exhibited more skill in forecasting RI in the east Pacific and North Atlantic (hereafter, Atlantic refers to the North Atlantic) basins[2]. However, the intensity forecast errors for RI events are still approximately 2–3 times larger than non-RI events, depending on forecast lead time[3]. The forecasting challenges associated with these RI events were likely exacerbated by the recent upward trend in the proportion of storms that achieved RI[4–10]. Projections for an increase in the probability of RI in the future[11,12] suggest that the forecasting of TCs could grow even more challenging.

Still, the anthropogenic contribution to the recent changes in TC intensification rates and the favorability of TC environments has received little attention. Various basin-specific forms of multidecadal variability (e.g., the Atlantic Multidecadal Variability [AMV] and the Pacific Decadal Oscillation [PDO]) becoming more conducive to TC development[7,13] has coincided with the relatively short period with reliable TC intensity estimates[14]. Multidecadal natural variability, anthropogenically-forced climate change, and observational data limitations need to be carefully considered in order to identify the various contributions to the recent trends. Bhatia et al.[6] first attempted to understand the role of natural climate variability in the increase of

[1]Guy Carpenter, New York, NY, USA. [2]National Centre for Atmospheric Science and Department of Meteorology, University of Reading, Reading, Berkshire, UK. [3]Department of Geosciences, Princeton University, Princeton, NJ, USA. [4]High Meadows Environmental Institute, Princeton University, Princeton, NJ, USA. [5]NOAA/Geophysical Fluid Dynamics Laboratory, Princeton, NJ, USA. [6]The Climate Service, an S&P Global company, Madison, WI, USA. [7]Englehart Commodities Trading Partners, London, UK. [8]NOAA/Geophysical Fluid Dynamics Laboratory, Princeton, and Engility Inc., Dover, NJ, USA. ✉e-mail: Kieran.bhatia@gmail.com

TC intensification by comparing observed trends (1982–2009) to trends from an unforced control simulation of the High-Resolution Forecast-Oriented Low Ocean Resolution model (HiFLOR[15]). HiFLOR is a high-resolution coupled global climate model that can recover many aspects of the highest TC intensification rates observed in nature and capture the connection between low-frequency climate oscillations and TC behavior[11,15,16]. The results suggested a detectable increase in Atlantic intensification rates with a positive contribution from anthropogenic forcing but required a longer time series to detect a robust trend at the global scale.

Here, we expand on the results of Bhatia et al.[6] by examining 24-h intensification rates during the 36-year period between 1982–2017 and comparing the intensification trends to those in HiFLOR. We then utilize reanalysis datasets to examine trends in the observed storm-ambient and tropical-mean environments during the same period. Finally, the observed trends in the tropical-mean environments are compared to those in Coupled Model Intercomparison Project phase 6 (CMIP6[17]) simulations.

## Results

### Comparing observational and model intensification trends

Following the work of Bhatia et al.[6], we calculate 24-h intensification trends using two observational datasets, the Advanced Dvorak Technique-Hurricane Satellite-B1 (ADT-HURSAT[18]) and International Best Track Archive for Climate Stewardship (IBTrACS[19]). Overall, ADT-HURSAT is considered a more reliable dataset for trend analysis because it is derived from a temporally and spatially homogenized record of TC intensity[18]. IBTrACS relies on the best-available observing practices from different operational agencies across the world, and thus provides more accurate intensity measurements for individual TCs. A detailed discussion of the strengths and weaknesses of each dataset is presented in Bhatia et al.[6].

Using IBTrACS and the updated ADT-HURSAT dataset[20], we apply intensity, longevity, and latitude thresholds (see Methods) to evaluate the intensification trends of TCs over the ocean between 1982–2017. RI ratio, defined as the number of 24-h intensity changes greater than 30 knots divided by all 24-h intensity changes, is selected as a normalized metric to capture how the probability of TC RI has evolved with time. Figure 1 shows annual RI ratio between 1982–2017 for the Atlantic, East Pacific, West Pacific, Australian, and South Pacific basins as well as global data. The Indian and Central Pacific basins are respectively excluded from the analysis because of well-documented gaps in satellite coverage in the early portion of the times series[18] and infrequent TCs limiting the sample size. The exclusion of these basins from the analysis does not significantly change any of our conclusions.

Throughout all basins, there are significant (rejecting the null hypothesis of no trend at the $p = 0.05$ significance level) upward trends in RI ratio defined using IBTrACS data (Fig. 1), which agrees well with recent studies[6,10]. The west Pacific, Atlantic, and Australian basins show significant upward trends in ADT-HURSAT and IBTrACS data and are largely responsible for the significant upward global trend in both observational datasets. The agreement between the two observational datasets and similar results from recent studies[10,20] suggests the trends in these basins are robust. The change in the proportion of intensity changes undergoing RI in these basins is part of an overall broadening of the intensity distribution and not just an increase in intensity changes greater than 30 knots. Supplementary Fig. 1 is formulated[21] using a similar methodology to Fig. 2 in Bhatia et al.[6] and shows that the majority of the upper quantiles are increasing and lower quantiles are decreasing, reflecting less storms maintaining a steady intensity.

In some basins, there are noteworthy discrepancies in the trends of RI ratio between the two observational datasets. In particular, the east Pacific and south Pacific show different signed RI ratio slopes between ADT-HURSAT and IBTrACS, with ADT-HURSAT showing negative slopes and IBTrACS showing positive slopes. Even with more

temporally-consistent observational data integrated into IBTrACS intensity estimation during the satellite era, trends calculated using IBTrACS data are likely overstated. For the remainder of the manuscript, we focus on the west Pacific, Atlantic, Australian, and global datasets because of the agreement in the sign of the trends in the observational datasets for these basins.

To assess the extent to which these trends in RI ratio can be explained by natural, internal climate variability, we follow a similar methodology to Bhatia et al.[6]. Internal natural variability in TC intensification is estimated based on the internal variability from simulations of preindustrial conditions modeled using HiFLOR. Specifically, we use HiFLOR simulations with anthropogenic forcing (e.g., CO2, aerosols, and ozone) and natural forcing (e.g., volcanic aerosol loading and solar insolation) held fixed at "pre-industrial" levels representative of the year 1860 (1860CTL; See Methods). To adjust for systematic errors in the HiFLOR distribution of intensification and ensure realistic slopes in RI ratio, quantile delta mapping (QDM)[22] is applied to each basin[6]. Overlapping 36-year RI ratio slopes in the bias-corrected 1860CTL are then compared to the observed ADT-HURSAT and IBTrACS RI ratio slopes between 1982 and 2017.

Figure 2 contains raincloud[23] plots that show the distribution of RI ratio slopes for the QDM-corrected 1860CTL. The observed slopes for IBTrACS and ADT-HURSAT during 1982–2017 are overlaid. In all the analyzed basins, the slope of the annual RI ratios for IBTrACS are above the 99th percentile of the slopes of the bias-corrected HiFLOR 1860CTL. The ADT-HURSAT slope is also significantly higher than, at the 95th percentile or above, the HiFLOR slopes for the Atlantic and West Pacific (Australian basin is at the 94th percentile) basins as well as globally. The emergence of global and basin-specific trends in ADT-HURSAT that are extremely rare or outside the range of internal climate variability simulated by HiFLOR increases the likelihood that the recent uptick in TC RI is an anthropogenically-forced trend. Furthermore, Fig. 6 of Bhatia et al.[6] supported this conclusion when they compared RI ratio in 1860CTL and the HiFLOR simulations with stronger anthropogenic forcing (1940CTL, 1990CTL, and 2015CTL). Using this figure, they showed that anthropogenic forcing significantly increases extreme TC intensification rates in the HiFLOR model.

HiFLOR's realistic simulation of TCs and their connection to climate variability instills some confidence in the significance of these results[11,15,16]. However, it is important to emphasize that these results rely on one bias-corrected climate model to estimate internal climate variability, and the model-dependence of the results should be explored when other coupled climate models can simulate RI statistics. Also, the horizontal atmospheric resolution ($0.25° × 0.25°$) in the model suggests that some of the small-scale processes associated with TC RI are not fully resolved[24,25]. A concurrent anthropogenically-forced increase in the conduciveness of storm environments would suggest that it is very likely that anthropogenic forcing contributed to the observed increase in RI ratio.

### Observed trends in the environmental favorability for tropical cyclone intensification

Recent studies have suggested that rising sea surface temperatures and potential intensities could provide a physical explanation for more intense TCs and extreme intensification events[7,10,18,26,27]. However, there is not yet analysis on whether changes in storm tracks and variability on weather time scales could potentially prevent TCs from experiencing these environments that are more conducive for RI[28,29]. In this section, we examine trends in the local environments surrounding TCs to determine whether intensification trends can be explained by storms experiencing more favorable environmental conditions.

Four environmental variables, documented as vital for TC RI[1], are calculated using the European Centre for Medium-Range Weather Forecasts (ECMWF) fifth-generation global atmospheric reanalysis (ERA5[30]) and Modern-Era Retrospective Analysis for Research and

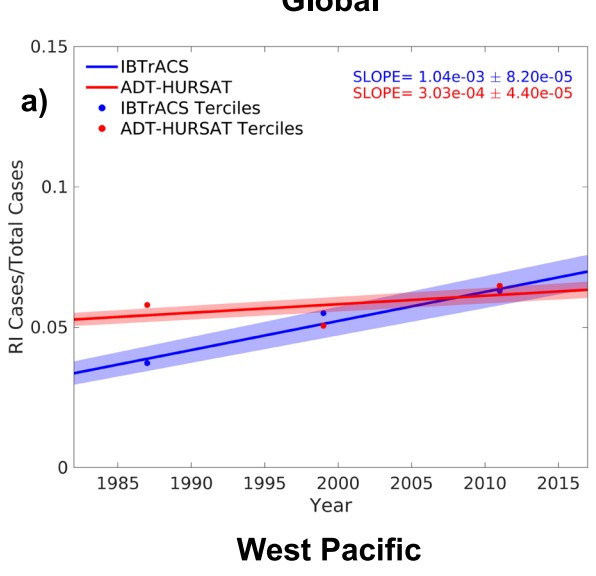

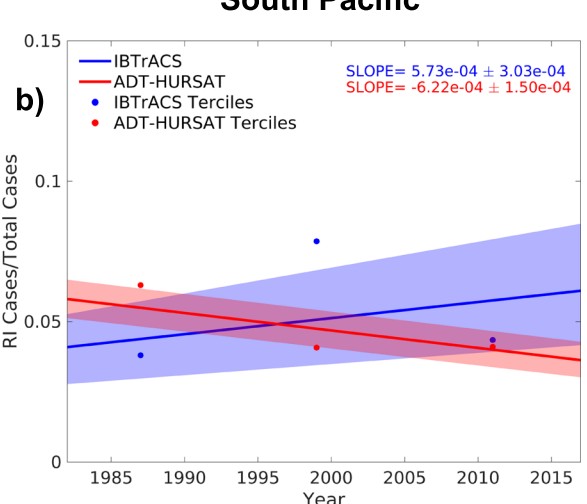

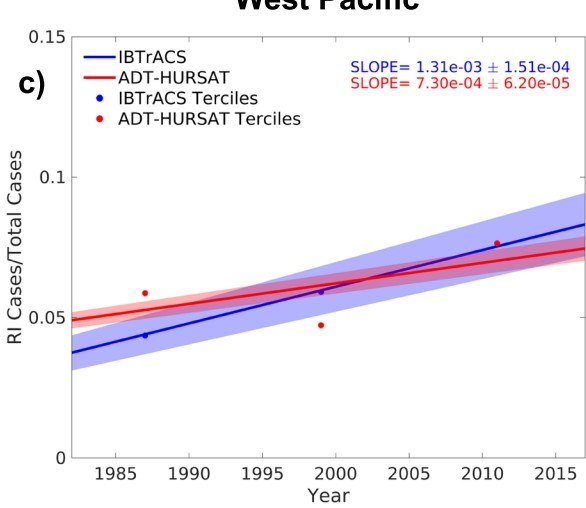

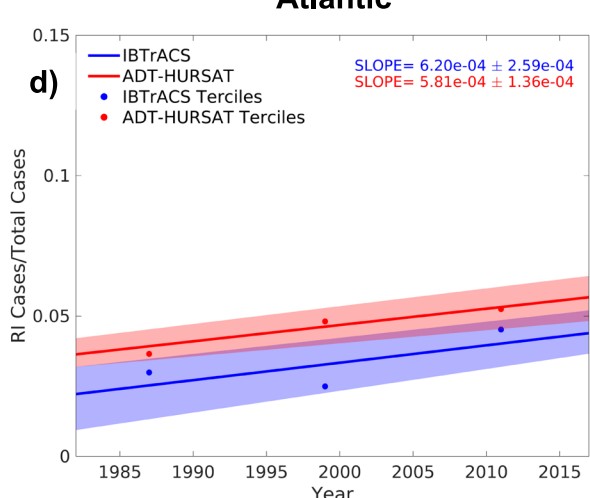

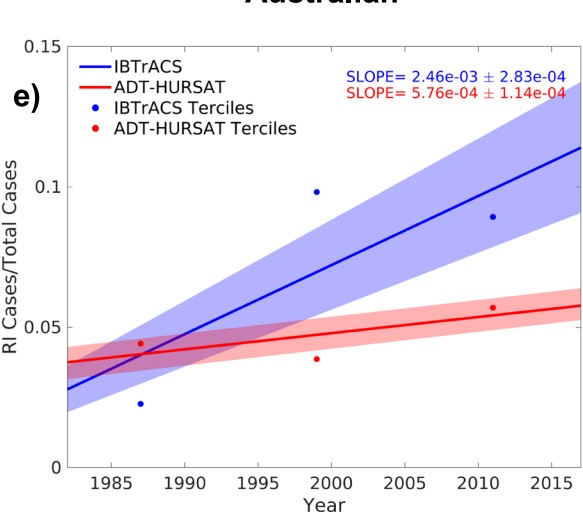

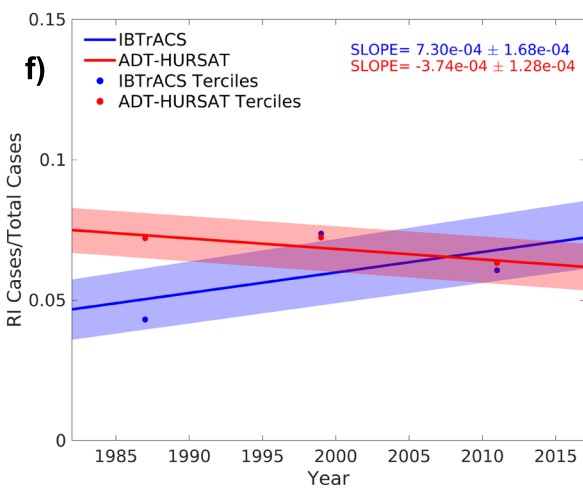

**Fig. 1 | Rapid intensification ratio trends.** Observed trends in the rapid intensi-fication (RI) ratio of ADT-HURSAT (blue) and IBTrACS (red) over the 36-year period 1982–2017 using (**a**) Global and (**b**) South Pacific (**c**) West Pacific (**d**) Atlantic (**e**) Australian (**f**) and East Pacific data. RI ratio is defined as the number of 24-h intensity changes above 30 knots divided by the total number of 24-h intensity changes. The 36-year period is divided into three terciles (1982–1993, 1994–2005, 2006–2017) and plotted as circles. The slopes of the trend lines and their 90 percent confidence intervals are respectively demarcated as solid lines and shading. The slopes and confidence intervals are calculated using 1000 randomly perturbed samples of the observational data. Shading represents the 5th and 95th percentiles of the 1000 regressions with these randomly perturbed observational data (Methods).

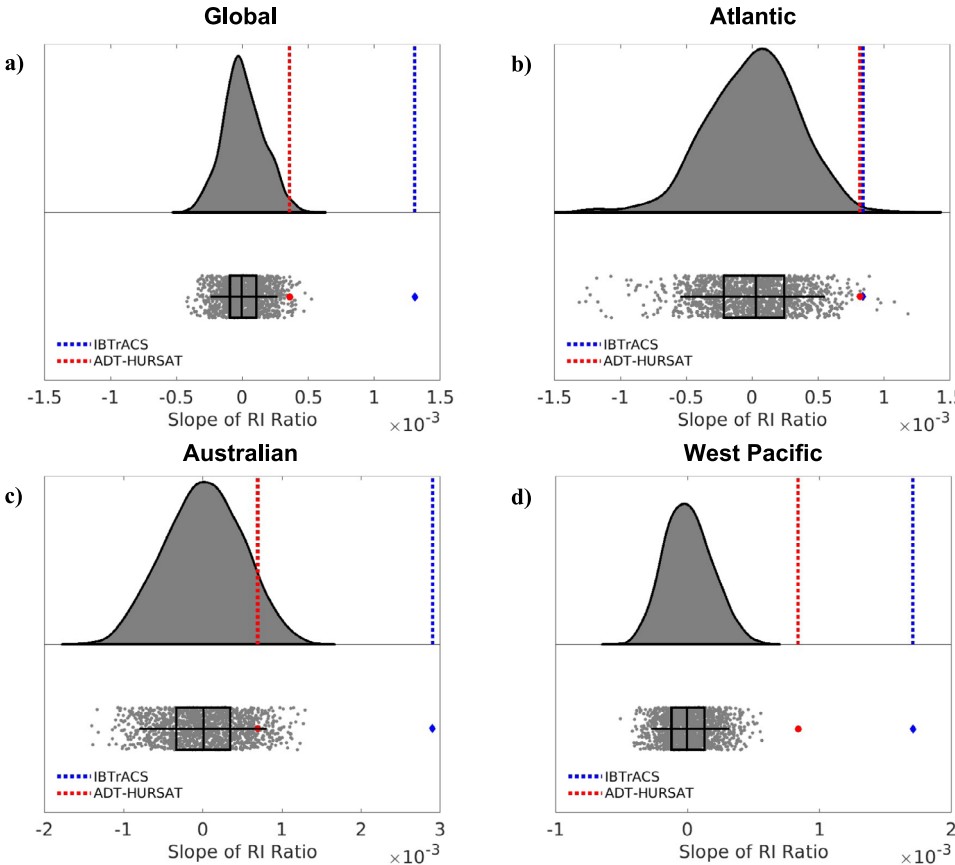

**Fig. 2 | Observed trends in rapid intensification ratio vs. HiFLOR natural variability.** Raincloud plots represent the distribution of rapid intensification (RI) ratio slopes in the Quantile Delta Mapping-corrected 1860 HiFLOR control simulation using a half-violin plot (top) and raw jittered data and box-and-whisker plots (bottom) for four basins (**a**) Global (**b**) Atlantic (**c**) Australian (**d**) West Pacific. Each of the HiFLOR slopes is calculated by applying least squares regression analysis to annual RI ratio values in 1,414 (number of available years reduced by 36) overlapping 36-year periods. The center line of the box represents the median and is bounded by the 25th and 75th percentiles of the data, and the whiskers bracket approximately 95% of the data. IBTrACS and ADT-HURSAT trends in annual mean RI ratio between 1982–2017 are respectively represented in bottom (top) subplot by a blue diamond (blue dotted line) and red circle (red dotted line).

Applications, version 2 (MERRA-2[31]): vertically-averaged relative humidity (RH; at 850, 700 and 600 hPa), vertical wind shear (SHR; between 850 and 200 hPa), sea-surface temperature (SST), and potential intensity (PI[32]). PI and SST are related metrics, but PI is uniquely impacted by the tropospheric profile of temperature and moisture. Calculated using the large-scale environment, PI represents an upper limit of TC intensity[33] that is derived from the thermodynamic disequilibrium between the surface of the ocean and the upper atmosphere[34]. When PI increases, the theoretical intensity range for a storm expands and greater 24-h intensity changes are possible. For these reasons and because of the ongoing research on the evolution of the relationship of PI and SST under climate change, we include both environmental variables in our analysis analysis[35]. The difference between PI and current TC intensity is also analyzed because it has a slightly higher correlation with intensification than solely PI[36].

To isolate the storm-ambient environment without the vortex signature, we first track storms in both reanalyses[37,38], spectrally filter the relevant environmental field, match environmental values to IBTrACS intensity fixes, and then compute spatial averages (further detail in Methods section). Before analyzing trends in the storm-local environments, it is important to demonstrate the viability of the selected ERA5 environmental variables at diagnosing situations favorable for TC RI. We primarily focus on ERA5 results because of its superior resolution and data-assimilation techniques[30]. Similar to Kaplan and DeMaria[1] Figs. 8 and 9, we compare the probability of RI above and below specific environmental thresholds. However, rather than taking the mean of the initial conditions for RI and non-RI cases to define the critical threshold for an environmental parameter, we solve the logit equation to attain the critical threshold that corresponds to the probability of RI in each basin. For example, the critical wind shear threshold in the Atlantic basin is 9.2 m/s which yields the mean probability between 1982–2017, 5.3%, of a 24-h wind speed exceeding 30 knots. Supplementary Table 1 includes the critical thresholds for the plotted basins and environmental parameters.

For each basin and parameter, Fig. 3a shows the RI ratio for cases satisfying the defined environmental thresholds as well as the RI ratio for cases where the threshold is not met. IBTrACS is exclusively used for the intensification data in this figure because we are not conducting trend analysis and therefore prioritize more accurate TC intensity measurements over the temporal consistency of observing capabilities. In all the basins, when the variables exceed (or in the case of shear, fall below) the defined thresholds, there is a significantly higher probability of RI compared to when the threshold is not met.

Figure 3b shows how the probability of RI changes as more critical storm-environment thresholds are exceeded. In every basin, the probability of TC environments satisfying multiple thresholds is low but when they occur, RI is more likely. For our given storm sample and intensity criteria (Methods), Supplementary Table 2 lists the number of times that the storm-local environments satisfied the different amounts of thresholds and how often RI occurred. Figure 3 reflects that TC intensity in the IBTrACS dataset evolves in a physically consistent way with the defined ERA5 environmental parameters and demonstrates that TC intensification is highly sensitive to changes in RH, SST, PI, and SHR. The documented relationship between TC

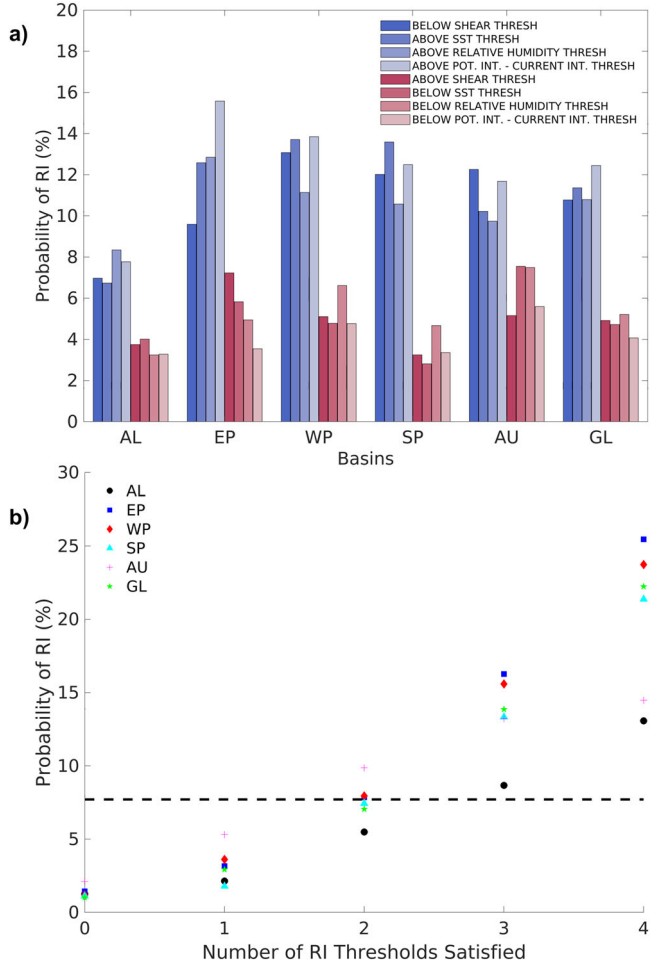

**Fig. 3 | Storm environment and rapid intensification probability. a** Histograms for six basins (Atlantic [AL], East Pacific [EP], West Pacific [WP], South Pacific [SP], Australian [AU], and Global [GL]) and four ERA5 environmental variables (Shear, Relative Humidity, Sea Surface Temperature, Potential Intensity-Current Intensity) showing the probability of rapid intensification (RI) for cases satisfying the critical environmental thresholds (blue histograms) and the probability of RI for cases where the critical thresholds are not met (red histograms). Critical thresholds are calculated by solving a logit equation for the environmental variable value that yields the average basin-wide probability of RI. **b** The probability of RI for six basins dependent on the number of critical thresholds (as defined in text) met. The black dashed line indicates the global probability of RI for all cases.

intensification and the storm environment is supported by theory and numerous modeling experiments[39–43]. Although internal dynamical processes of TCs are also important for RI[44], they are challenging to quantify and predict even with high-resolution numerical weather prediction models[3]. Additionally, it is not currently possible to assess how the internal dynamical processes will change with global warming using global climate models while large-scale environmental conditions are well-observed and captured by global climate models. Hence, we aim to develop understanding on how the environmental conditions surrounding TCs, rather than TCs' mesoscale processes, respond to a changing climate.

To determine if these unique synoptic situations conducive to RI are becoming more probable, we examine the annual proportion of cases meeting 3 or 4 thresholds and 0 or 1 threshold. Figure 4 shows how these environmental proxies for RI favorability evolve between 1982–2017 in the Atlantic, West Pacific, Australian, and global datasets. Excluding the Australian basin, there is a significant ($p < 0.05$ using the Wald Test[45]) increase in the proportion of fixes where 3 or 4 thresholds

are met and a significant decrease in the proportion of fixes where 0 or 1 threshold is met. The Atlantic basin displays the largest changes in environmental favorability for RI during the 36-year period with more than a doubling in the annual proportion of cases satisfying 3 or 4 thresholds and a greater than 50% reduction in the annual proportion satisfying 0 or 1 threshold.

Supplementary Figs. 2 and 3 provide additional evidence of local TC environments becoming more favorable. In Supplementary Fig. 2, the 75th and 95th percentiles of global RH, SST, PI, and SHR are plotted for 1982–2017. Both PI and SST show the most robust changes during the 36-year period which highlights the improved thermodynamic situations surrounding storms. Supplementary Fig. 3 further explores the time evolution of the local thermodynamic environments around storms in the Australian, Atlantic, West Pacific, and global datasets. In Supplementary Fig. 3, the probability of RI is contoured based on a logistic regression with two predictors, SST and PI, and the mean values of SST and PI for 1982–1993, 1994–2005, and 2006–2017 are plotted as red plus signs. For all the plotted basins (besides the last two terciles in the Australian basin), the later terciles progressively shift to higher SSTs and PIs and thus move across the contours to environments that are more favorable to RI. Supplementary Figs. 2 and 3 support the trends captured in Fig. 4 and explain why more storm-local environments are satisfying the key thresholds later in the time series. Additionally, the trends in Fig. 1 and Supplementary Fig. 1 align well with theory that the distribution of TC intensity and intensification should shift to higher values with more conducive thermodynamic environments[20,46].

## Anthropogenic influence and tropical-mean environments

Thus far, our analysis of the trends in TC environments have focused on storm-local, 6-hourly data. This granular analysis is crucial to better understanding the changes in the most relevant temporal and spatial scales for the intensification of individual storms. However, most climate models cannot resolve the temporal and spatial scales necessary to capture TC RI or the influence of storm-local environments on TC RI. In this section, we take spatial averages of synoptic variables in tropical ocean regions ("tropical-mean") as a proxy for TC RI in global climate models and check whether the trends in storm-local environments manifest in the tropical-mean values.

Figure 5 shows the annual trend in tropical-mean (defined in Methods) RH, SST, PI, and SHR in the MERRA-2 and ERA5 datasets. The plotted environmental fields represent "peak TC-season" averages in both hemispheres: August-September-October in the northern hemisphere and February-March-April in the southern hemisphere. The slopes of the different environmental variables closely resemble those calculated with storm-local environmental data. Significant upward trends ($p < 0.05$ using the Wald Test) are observed for PI and SST in both hemispheres and reanalyses, while shear exhibits a small but insignificant downward trend in both hemispheres and reanalyses. RH has opposite-signed trends depending on the reanalysis dataset (positive in ERA5 and negative in MERRA-2) and there is a large disagreement in the annual-mean values in the two reanalyses: a potential consequence of their differing data-assimilation techniques.

To assess the influence of anthropogenic forcing (including aerosols and greenhouse gasses) on tropical-mean environments, we compare multiple CMIP6 simulations using different forcing estimates. The all-forcing simulations (AllForc) include both anthropogenic forcing as well as natural forcing from volcanoes and solar variability. The greenhouse gas only forcing (GHGforc) and natural only forcing (Natforc) CMIP6 simulations use subsets of the AllForc simulations. We calculate the linear trend in peak-season SST, PI, SHR, and RH over the period 1982–2014 (2015–2017 is not available) for each ensemble member to create two types of plots in Fig. 6. Supplementary Table 3 contains the list of available CMIP6 simulations for each variable, and

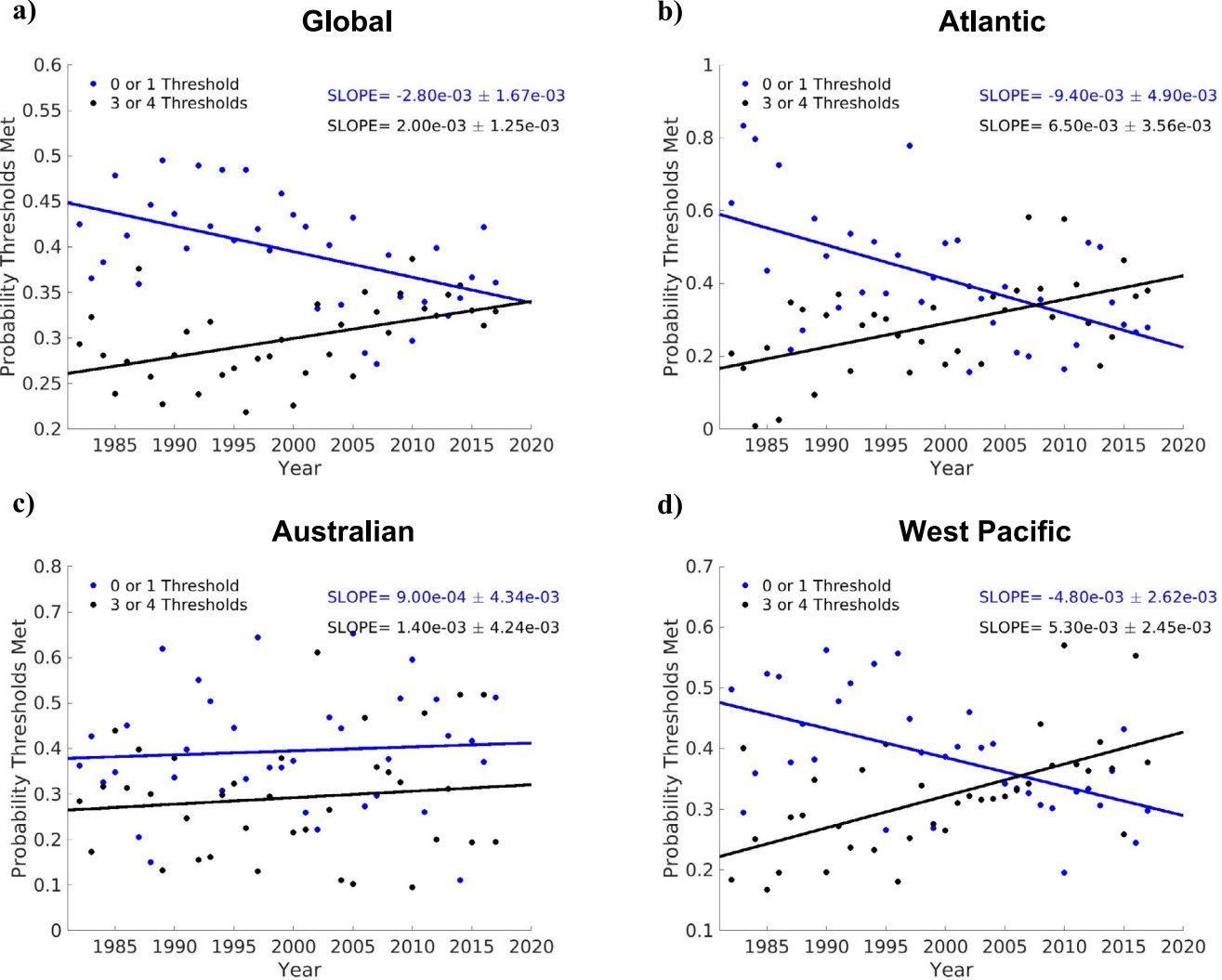

**Fig. 4 | Observed trends in the favorability of storm environments.** Observed trends in the annual probability of satisfying 3 or 4 (black) and 0 or 1 (blue) critical ERA5 environmental thresholds over the 36-year period 1982–2017 using (**a**) Global and (**b**) Atlantic (**c**) Australian (**d**) West Pacific data. Annual values are denoted by dots, and the slope derived from least squares regression of annual values are plotted as solid lines. The slope of both lines and their 95% confidence interval (Wald Test) is shown in the top right corner of each subplot.

the detailed methodology to prepare this figure is summarized in our Methods section. Figure 6a, c, e, g respectively show the probability density function (pdf) of the 1982–2014 slopes of all the ensemble members for SST, PI, RH, and SHR. The observed MERRA-2 and ERA5 trends are overlaid on the plots for comparison. Figure 6b, d, f, h respectively show the annual mean values of SST, PI, RH, SHR for the equally-weighted ensembles. The equally-weighted ensembles are constructed by normalizing each ensemble member by the total number of members for a modeling center and helps mitigate the outsized influence of models (e.g., CanESM5) with significantly more ensemble members.

Pairwise *t*-tests and Kolmogorov–Smirnov tests[47] are applied to compare the slope distributions in Fig. 6a, c, e, g. The title of each pair of plots for an environmental variable denotes the *p* values of these statistical tests comparing the CMIP6 ensemble slopes. For all variables, the tests reveal the mean of the AllForc and NatForc slopes are significantly different from one another, and the two samples are likely drawn from different probability distributions. Additionally, a student's *t*-test[48] is used to compare the slopes of the equally-weighted AllForc and NatForc ensembles for SST and PI in Fig. 6b, d, f, h. The AllForc equally-weighted ensemble mean slopes for SST and PI are significantly different from zero and those produced by NatForc.

Statistical tests yield similar results for RH but the equally-weighted ensemble mean trend in SHR for the historical simulations is not significantly different from zero or from the NatForc simulation.

The 1982–2014 trends in MERRA-2 and ERA5 SST are located near the middle of the CMIP6 AllForc pdf but are outside the NatForc pdf. These results indicate a detectable anthropogenic influence on SST in these regions. In the case of PI, ERA5 and MERRA-2 trends are again outside of the pdf of the natural forcing slopes but are also outside of the pdf of the AllForc ensemble. The magnitude of the observed trend (-1 m/s/year) in the reanalyses (stars in panel c) is more than 3 times larger than the mean slope (-0.3 m/s/year) of the equally-weighted AllForc ensemble (gray line in panel d). This result constitutes a detectable but largely unexplained trend in the MERRA-2 and ERA5 PI time series. However, the MERRA-2 and ERA5 trends are in the same direction as the anthropogenically forced signal (inferred by comparing the AllForc and NatForc pdfs), suggesting that anthropogenic forcing is likely contributing to the observed changes in PI.

Further research is required to better understand the source of the discrepancy between the PI trends in the AllForc ensemble and the reanalyses, but initial analysis suggests the discrepancy in the vertical structure of temperature changes (ERA5/MERRA-2 vs. CMIP6) is important for the divergent tropical PI trend behavior. A recent study

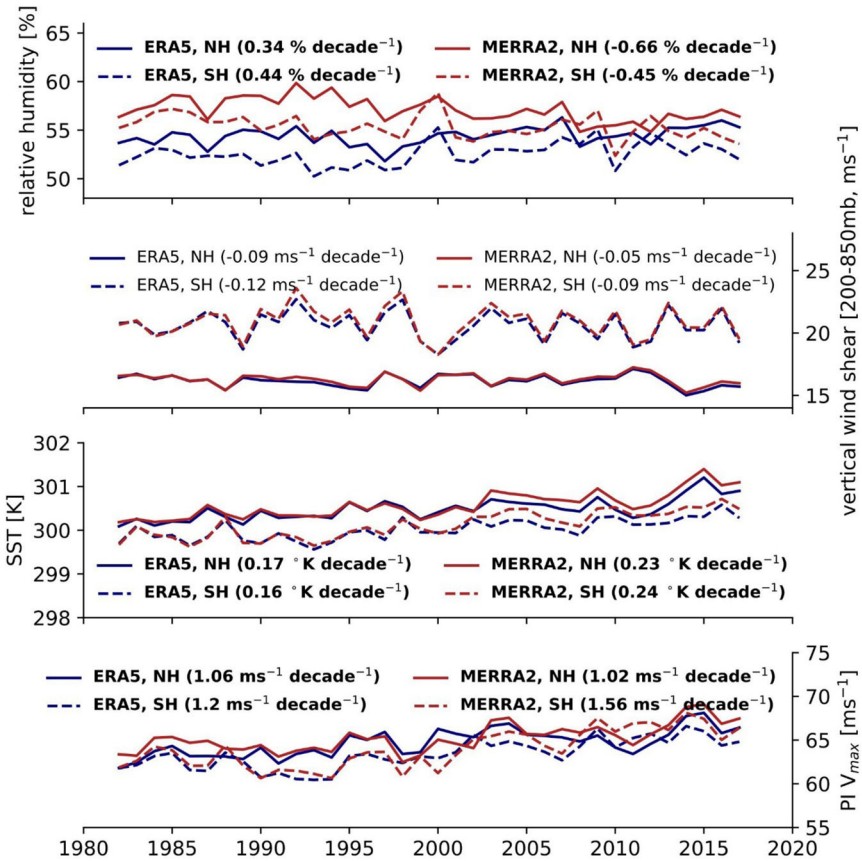

**Fig. 5 | Trends in the favorability of tropical-mean environments.** Observed trends in tropical-mean relative humidity (RH), vertical wind shear (SHR), sea surface temperature (SST), and potential intensity (PI Vmax) (top to bottom) in MERRA-2 (red) and ERA5 (blue) between 1982–2017. Dashed lines are used to connect annual southern hemisphere tropical-mean values and solid lines are used to connect annual northern hemisphere tropical-mean values. The slopes of each reanalysis dataset and hemisphere combination are included in the legends of each subplot, and the text is in bold if the slope is significant (Wald test, $p < 0.05$).

by Keil et al.[49] found that climate models on average overestimate the amount of recent (1979–2014) upper-tropospheric warming for a given lower-tropospheric warming. They suggested that the unrealistically large upper tropospheric warming in CMIP6 models was caused partly by imperfect convective parametrizations spuriously increasing latent heat release in the upper troposphere. Supplementary Fig. 4 displays trends in the tropical vertical temperature profiles in CMIP6, MERRA-2, and ERA5 between 1982–2014. The discrepancy between the trends in the upper troposphere appears to support Keil et al.[49] and partially explain the differing trends in CMIP6 and the reanalysis data. This finding is particularly meaningful and demands further research because it suggests that future increases of TC intensities and RI could be underestimated by current climate model-based projections that contain this bias.

## Discussion

This study leverages new datasets, climate model output, and analysis techniques to explicitly examine the question of whether climate change has contributed to the observed changes in TC intensification and environments surrounding TCs. Over the 36-year period 1982–2017, we observed a robust global increase in the proportion of RI events in a spatially and temporally homogeneous dataset (ADT-HURSAT) which was significantly ($p < 0.05$) higher than the trends in HiFLOR. The emergence of a significant trend in any TC metric at the global scale is noteworthy because it is unclear whether natural climate variability can modulate TC behavior at this spatial scale. The increase in the probability of RI is supported by storm-local and tropical-mean environments becoming more favorable to intensification.

Higher SSTs in recent years are primarily caused by anthropogenic drivers, and anthropogenic forcing has also significantly contributed to recent increases in tropical-mean PI. These environmental changes manifest in both coarse and granular temporal and spatial scales which suggests that track variability (such as a shift to more poleward locations[50]) is unlikely to prevent additional increases in TC intensification with further anthropogenic warming. Given the high confidence in growing atmospheric greenhouse gas concentrations over the next two decades and implications for continued SST warming, bolstering the development of high-resolution models capable of better resolving mesoscale processes and the environments surrounding TCs should be a societal priority. Additionally, our conclusions indicate that anthropogenically-forced climate change has already contributed to the observed, detectable increase in the proportion of rapidly intensifying hurricanes and highlight the immediate need to improve coastal resilience to prepare against these dangerous events.

## Methods
### Tropical cyclone observations
We used the International Best Track Archive for Climate Stewardship (IBTrACS[19]), v04r00, and the Advance Dvorak Technique-Hurricane Satellite-B1 (ADT-HURSAT[18,20]) for the period 1982–2017. The ADT-HURSAT data record was recently expanded by 8 years to span the 39-year period between 1979–2017. We omit the first three years of the record where limited geostationary data results in missing storms and focus on the 36-year period between 1982–2017. For our IBTrACS analysis, we only consider best-track data from the National Hurricane Center for the Atlantic and east

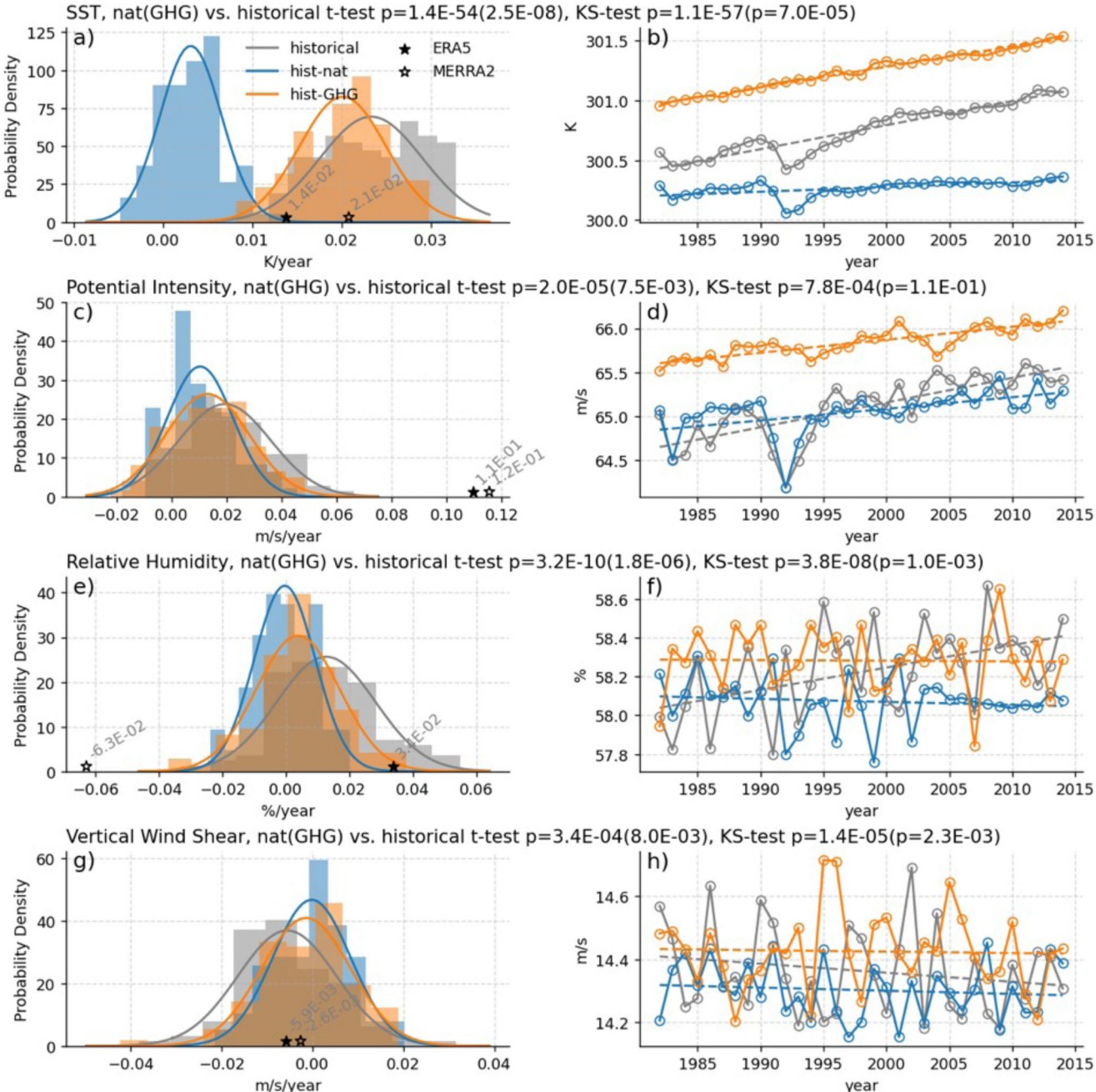

**Fig. 6 | Comparing tropical-mean trends in CMIP6 simulations. a–h** Observed trends between 1982–2014 in tropical-mean sea surface temperature (SST), potential intensity (PI), relative humidity (RH), and vertical wind shear (SHR) for CMIP6 historical (gray), hist-nat (blue), and hist-GHG (orange) simulations. **a**, **c**, **e**, and **g** show probability distribution functions (pdfs) of the tropical-mean slopes of all the available ensemble members for each simulation type. The pdf curves are calculated using a maximum likelihood estimation of the normal distribution parameters. The slopes of the MERRA-2 and ERA5 tropical-mean values for all four environmental variables are calculated between 1982–2014 and illustrated as black stars on the subplots. **b**, **d**, **f**, and **h** show the annual values of the tropical-mean environmental variables derived from the equally-weighted ensemble mean of historical, hist-nat, and hist-GHG simulations. The title of each pair of plots for an environmental variable denotes the $p$ values of a pairwise $t$ test and Kolmogorov–Smirnov test comparing historical and hist-nat (hist-GHG) ensemble slopes in **a**, **c**, **e**, and **g**.

Pacific and the Joint Typhoon Warning Center for the remainder of the globe[19]. One of the benefits of only using data from these U.S. agencies is they follow the same definition of maximum winds: the highest 1-min average at 10 m height over a smooth surface[51]. Best-track data start as operational estimates of the intensity and track of a TC and are refined at the end of a TC season with a combination of in situ (e.g., dropsondes, scatterometers, buoys), radar, and satellite measurements. Best-track intensity and position estimates are available every six hours at the four synoptic times

(0000, 0600, 1200, and 1800 UTC) and are recorded to the nearest 5 knots (1 kt = 0.5144 m s⁻¹) and 0.1° latitude/longitude[52].

The creation of ADT-HURSAT consists of three main steps. Geostationary satellite imagery is first analyzed from International Satellite Cloud Climatology Project (ISCCP)-B1 data[53–55]. Then, the data is centered on IBTrACS TCs and subsampled to be both spatially and temporally homogeneous. Finally, a simplified version of the advanced Dvorak technique[56] is used to evaluate the data and determine a maximum TC wind speed. ADT-HURSAT data are produced every three

hours based on satellite data that has been uniformly subsampled to a horizontal resolution of 8 km, and wind speeds are recorded to the nearest tenth of a Dvorak "T-number" (depending on the current intensity, between 1–3 knots).

### Defining storm-local and tropical-mean environments

Data for the analysis of cyclone environments were taken from ERA5[30] and MERRA2[31,57]; reanalyses. Reanalyses combine a forecast model with continuous assimilation of observational data to construct a global representation of historical atmospheric variability. The native spectral resolution of ERA5 is T639 (nominally 31 km) and the native resolution of MERRA-2 is $0.5° × 0.625°$ (nominally 50 km). We focus on these reanalyses because they capture realistic moisture, temperature, and wind values in the lower atmosphere[58,59]. For storm-local environment calculations, we identified and tracked tropical cyclones in ERA5 and MERRA-2 objectively using the Lagrangian feature-tracking algorithm−TRACK−of Hodges[37]. This methodology was documented in greater detail by Hodges et al.[60].

Vertically averaged relative humidity (at 850, 700 and 600 hPa), vertical wind shear (computed as the square-root of the sum of the squared differences in the zonal and meridional winds between 850 and 200 hPa), and sea-surface temperature were spectrally filtered to T11 resolution to remove cyclonic circulation features and retain only the large-scale, background environmental fields. Sensitivity analysis of ERA5, the highest-resolution reanalysis available, shows that 95% of the T639-resolution cyclonic circulation is removed at a T11 truncation (Supplementary Fig. 5). Along-track sampling of mean values from the spectrally filtered fields within a 5° storm-centered radius (geodesic) was performed. To compute potential intensity along cyclone tracks, reanalysis data (sea-surface temperature, mean sea-level pressure, air temperature, and specific humidity) were regridded first to 1° resolution. Regridding from the native reanalysis resolutions to 1° has a negligible effect on potential intensity and no spectral filtering was performed. Potential intensity was computed by taking input fields from the grid cell nearest to the storm center at each timestep, following Bister and Emanuel[61] and using published code[62,63] (Gilford et al.). Vertical soundings of temperature and specific humidity were constructed from reanalysis data on 14 isobaric levels (1000, 925, 850, 700, 600, 500, 400, 300, 250, 200, 150, 100, 70, and 50 hPa). By default, the code allows reversible ascent and dissipative heating; the ratio of the exchange coefficients of enthalpy and momentum flux is 0.9; and output velocity is scaled by 0.8 to reflect surface drag. Further discussion of these constants is given in Emanuel[33] and Gilford et al.[62,63].

The tropical-mean of these fields are also calculated but with no spectral filtering. Annual averages are comprised of peak-TC-season means in each hemisphere, August-October (ASO) in the Northern Hemisphere and February-April (FMA) in the Southern Hemisphere and then averaged between the two Hemispheres (area-weighted). June-October (JJASO) in the northern hemisphere and December-February (DJFMA) were also tested but yielded comparable results. In the northern hemisphere, the tropical-mean is an area-average of environmental values over the ocean between 10°−30°N and 40°E-20°W. In the southern hemisphere, the bounds for averaging are 10°−30°S and 30°E-150°W.

### HiFLOR experiments

HiFLOR control simulations introduced in Murakami et al.[64] and Bhatia et al.[6] were used here to represent natural (unforced) climate variability and provide the framework for exploring anthropogenic effects on TC intensification. We focused on the control simulation that used anthropogenic forcing fixed at 1860 (1860CTL) levels because it has the longest simulation length: 1500 years. The first 50 years of the simulation were disregarded

to mitigate effects of model drift. The approach developed by Harris et al.[65] is used to track TCs in HiFLOR and is applied using the parameter values of Zhang et al.[16] and Murakami et al.[15]. The warm core criteria discussed in Murakami et al.[15] is applied to the HiFLOR data before analysis.

### CMIP6 experiments

We examine linear trends of the four tropical-mean environmental fields from all available CMIP6 simulations over the period 1982–2014. The fields are defined in an identical way to the observed fields in MERRA-2 and ERA5. Models and the number of ensemble members for each model that contain the relevant environmental variables are listed in Supplementary Table 3.

### Storm criteria

For consistency, intensity change values in HiFLOR, and the observational datasets are rounded to the nearest five knots. We only consider TCs that are active for at least 72 h and exceed wind speeds of 34 knots for at least 36 h. We restrict our analysis sample to only consider cases where the TC center is located over the ocean, the starting and ending TC position are below 40 degrees of latitude, and the TC intensity stays above 34 knots. We do not examine TC intensity changes above 40° latitude because storms typically complete extratropical transition[66] and achieve their lifetime maximum intensity equatorward of this latitude[50]. The warm core criteria discussed in Murakami et al.[15] are also applied to the HiFLOR data before analysis. For the analysis involving storm-local environments, all storm fixes within 0.5 degrees of land in any direction are removed to reduce the number of spurious PI readings.

### Uncertainty quantification

For Fig. 1 and Supplementary Fig. 1, we use Monte Carlo techniques to create random noise before analyzing the discretized data. Random noise prevents multiple data points from having the same value and provides an estimate of the typical error associated with measuring TC intensity. 1000 subsamples were produced by adding random noise from a uniform distribution on the interval $± 2× \sqrt{50}$ knots to each intensity change value. The magnitude of this random noise is derived by adding 5 knots of error in quadrature (propagation of errors stemming from the intensity change calculation), which is a conservative estimate for the typical error associated with each TC intensity observation. Figure 1 and Supplementary Fig. 1 involved the calculation of the 5th and 95th percentile in each of the 1000 subsamples for each year. 1000 slopes of each percentile or quantile were calculated, and the mean of the slopes was considered the best estimate of the 1982–2017 slope of the percentile or quantile. The 5th and 95th percentiles of the 1000 slopes were shaded as the uncertainty bounds.

## Data availability

Besides the HiFLOR data, all the data used for this study are publicly available without access codes. CMIP6 model output is available at https://esgf-node.llnl.gov/projects/cmip6/. The ERA5 data can be downloaded here: https://cds.climate.copernicus.eu/#!/search?text=ERA5&type=dataset. The IBTrACS data can be downloaded here: https://www.ncei.noaa.gov/products/international-best-track-archive. The ADT-HURSAT data can also be downloaded[20].

## Code availability

The code that supports the findings of this study is available from the corresponding author on request. The code used for tracking TCs in ERA5 and MERRA-2 are available here: https://gitlab.act.reading.ac.uk/track/track. The source code of the HiFLOR model can be found at https://www.gfdl.noaa.gov/cm2-5-and-flor.

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

## Acknowledgements

The authors thank Dr. Shuai Wang and Pier Luigi Vidale for their suggestions and comments during the internal review process. This work was supported by the Carbon Mitigation Initiative at Princeton University, and Award NA18OAR4320123 from the National Oceanic and Atmospheric Administration, U.S. Department of Commerce. The statements, findings, conclusions, and recommendations are those of the author(s) and do not necessarily reflect the views of the National Oceanic and Atmospheric Administration, or the U.S. Department of Commerce.

## Author contributions

K.B. designed the study, completed a majority of the analysis, and wrote the entirety of the manuscript. A.B. created Fig. 5 and Supplementary Fig. 5. A.B. performed tracking of ERA5, produced the storm-local and tropical-mean reanalysis environments for trend analysis. W.Y. created Fig. 6 and conducted the analysis of the CMIP6 tropical-mean environmental trends. K.H. developed the tracking algorithm to pair the environmental data to TC fixes and performed tracking of MERRA-2. G.V. helped develop the HiFLOR model and carry out the experiments. G.V. also provided key suggestions on research topics to explore. T.K. provided guidance on the analysis. H.M. provided key suggestions on the analysis and helped carry out the experiments. J.K. was one the developers of ADT-HURSAT data and provided it for analysis. K.D. helped design and write a description of the bias corrections for HiFLOR. C.W. was the primary developer of the bias corrections for HiFLOR. B.B. conducted early analysis of the CMIP6 tropical-mean environmental trends and supported the development of comparison metrics.

## Competing interests

The authors declare no competing interests.
