## [Peer Review File · Nature Communications]

A Potential Explanation for the Global Increase in Tropical Cyclone Rapid IntensificationREVIEWER COMMENTS

Reviewer #1 (Remarks to the Author):

Comments on "Tropical Cyclone Rapid Intensification: An Explanation for the Global Increase"

The paper compared the observed trends in TC intensification and TC environmental parameters to simulated natural variability in a high-resolution global climate model to identify the contribution from anthropogenic forcing. Physical explanations on increased TC RI events are by all means important. The findings in the paper will contribute to studies on TC RI in the future.

Here are two major concerns about the paper.

1. RI is a statistical concept, which is defined as approximately the 95th percentile of 24-h intensity changes. RI was suggested to be more likely just the tail of a distribution and not a distinct physical process (Kowch & Emanuel, 2015). The authors' previous study (Bhatia et al. 2019) found TC intensification rate increased during the period 1982-2009. Both negative slopes for the lower percentiles and positive slopes for the upper quantiles in the quantile regression were found in Bhatia et al. 2019 (Figure 2), which suggested a broadening of the intensity change distribution. Due to the broadening of the intensity change distribution, the 95th percentile of 24-h intensity changes increases from 30 knots to a higher rate, for instance, 35 knots. Therefore, the increased RI ratio, which was defined as the number of 24-hour intensity changes greater than 30 knots divided by all 24-hour intensity changes in the paper, suggests less storms maintaining a steady intensity and more storms exhibiting intensification rate higher than 30 knot. The possible influences of changing thermodynamic environments on less storms maintaining a steady intensity and more storm exhibiting high intensification rate should be discussed.

2. The paper applied the same method as Kaplan and DeMaria (2003) to identify the environmental thresholds for TC RI. RI cases in Kaplan and DeMaria (2003) were relative to non-RI cases, which included weakening and neutral storms. It is sure that some environmental conditions are different between RI and weakening/neutral storms. However, Hendricks et al. (2010) found that the environment of RI TCs and intensifying TCs is quite similar. Their study suggested, given the environment is favorable, the rate of intensification is not critical dependent on the environmental conditions and RI is mostly controlled by internal dynamical processes. The authors need to discuss the discrepancy.

Reference

Bhatia, K. T. et al. Recent increases in tropical cyclone intensification rates. *Nature Communications* 10, 635, doi:10.1038/s41467-019-08471-z (2019). 7

Hendricks, E. A. M. S., Peng, B. F., & Li, T. (2010). Quantifying environmental control on tropical cyclone intensity change. *Monthly Weather Review*, 138(8), 3243–3271. <https://doi.org/10.1175/2010MWR3185.1>

Kaplan, J. & DeMaria, M. Large-Scale Characteristics of Rapidly Intensifying Tropical Cyclones in the North Atlantic Basin. *Weather and Forecasting* 18, 1093-1108, doi:10.1175/1520-0434(2003)018<1093:lcorit>2.0.co;2 (2003).

Kowch, R., & Emanuel, K. (2015). Are special processes at work in the rapid intensification of tropical cyclones? *Monthly Weather Review*, 143(3), 878–882. <https://doi.org/10.1175/MWR-D-14-00360.1>

Reviewer #2 (Remarks to the Author):

In this work the authors analyse the occurrence of the largest tropical cyclone intensification rates in the observational datasets of the last decades and show that their increase is mainly linked to the increase in sea surface temperature and that such increase has likely been affected by anthropic activities. The study is well designed and the analysis generally supports the conclusions. The manuscript is very well written and includes the relevant details. However, in the present form it does not add much information to what we already know about tropical cyclone and anthropogenic climate change.

The fact that SST is a fundamental driver of tropical cyclone intensification is a well known thing that dates back to the work by Malkus and Riehl, 1960. More recently, Xu et al. 2016 and Xu and Wang 2018 showed the existence of a maximum potential intensification rate that primarily depends on SST. It is also well known that SSTs have been increasing over the last decades and that anthropic activities play role in this increase. The novelty in this manuscript is that not only TC intensification rate but also rapid intensification events are increasing, primarily driven by the increasing SSTs. The scientific literature has already highlighted the existence of the upward trend in extreme rapid intensification events over the last 30 years (Klotzbach et al. 2022).

In order to put the results presented in this manuscript in perspective, we need to consider the definition of rapid intensification events that is used, i.e. the number of events with 24hr intensity change larger than a fixed threshold. Considering that intensification rates have increased, mainly driven by higher SSTs and, related to SSTs, higher potential intensity, it is expected that all the intensification rates, including the higher ones, shifted to larger values. The authors are well aware of this expectation, and in fact they state that what is missing in the literature is an analysis on whether changes in storm tracks and variability on weather time scales could prevent TCs from experiencing large scale environmental conditions, such as the increased SST, that are more conducive for RI (lines 165-167). Their conclusion is that this did not happen. However, they did not fully investigate the issue: It would be interesting to know whether the distribution of intensification rate changed shape or not, whether the high end tail got higher or lower. In short, are data consistent with a shift of the intensification rate pdf to larger values or not? I encourage the authors to assess this.

It is acknowledged the fact that forecast errors for Rapid Intensification (RI) events are 2-3 times larger than for non-RI events, which indicates that some factors that favour their occurrence have not been identified yet, but this paper does not shed light onto this. Actually, the fact that the forecast error for RI events is large indicates that the large scale conditions that are known to favour their occurrence (and that are analysed in this manuscript) do not capture the key elements that lead to RI events. This should be highlighted.

Minor comments:

Rapid Intensification is identified as the 24hr intensity change above the 95th percentile on line 66 and above 30 knots on line 109.

Fig.1 : The shading in the fig comes from the perturbations added to the intensities before computing intensification rates, but I don't see how is the statistical significance of the slope evaluated. In other words, is the null hypothesis of no trend rejected? Please add some description of those details.

Selection of tracks: The analysis is restricted to TC with track starting and ending position below 40° of latitude (line 396). The authors conclude that track variability (such as a shift toward poleward locations) is unlikely to prevent additional increases in TC intensification with further anthropogenic warming (lines 307-309). Such conclusion however might not be supported by the present analysis, as the authors are not considering TC that end at latitudes higher than a fixed threshold. In theory, TCs

with tracks extending at higher latitudes could actually experience a decrease in the occurrence of rapid intensification events. Although I don't expect the results to significantly change, the inclusion of such storm seems to me necessary, in light of the goal of the study.

References:

Klotzbach et al. 2022, doi: 10.1029/2021GL095774

Malkus, J. S., and H. Riehl, 1960, doi:10.1111/j.2153-3490.1960.tb01279.x.

Xu et al. 2016 doi: 10.1175/JAS-D-16-0164.1

Xu and Wang 2018 doi: 10.1175/WAF-D-17-0095.1

Reviewer #3 (Remarks to the Author):

Dear Authors:

I read your manuscript "Tropical Cyclone Rapid Intensification: An Explanation for the Global Increase" with great interest. Overall, I found this paper to be a convincing expansion of Bhatia et al 2019, which took a thorough and sound methodological approach to address two key questions:

1. To what extent have rapid intensification events been increasing globally over the last 35 years?
2. What are the root causes of any increases?

Question 1 was addressed using the two leading datasets of TC climatology, one operational and one satellite-derived, and question 2 was addressed by demonstrating that key spatially averaged TC environmental metrics are showing trends in reanalysis far beyond the natural forcing variability in a suite of climate models.

Both questions were satisfactorily address in my opinion, with only a couple of points of clarification for the benefit of the reader and minor wording changes requested. Given the societal import of the research topic and the novelty of the treatment here, I recommend publication of this manuscript without delay after these minor issues are addressed.

General comments

1. Lines 161-179: I would like to see more discussion of what PI is both conceptually and physically when it is introduced, especially as SST and PI are both used as environmental proxies for the remainder of the paper, and of course there is a dependency in play there. I don't take any issues with the methodology of the study, but I do think some additional context in the introduction of the concepts would be beneficial in order to make the link between anthropogenic influence and RI outcomes clearer.
2. Lines 275-297: I think more needs to be said and shown here beyond "further research is required" to address the major differences between PI trends in the climate models and reanalyses. The physical explanation offered in the second paragraph seems highly credible, but I believe a supplemental figure comparing average tropical temperatures with height between the reanalyses and CMIP6 would go a long way for the reader in making it convincing. As is, that paragraph seems needlessly speculative.

Specific comments

1. Line 46: I would clarify that "worst forecasts" refers to high intensity errors at relatively short operational lead times.
2. Line 71-72: Add "intensity" before "forecast errors."
3. Lines 103/125: These lines repeat each other.
4. Line 158: Might be worth citing Walsh et al 2013 here?

5. Line 289: Something went wrong with citations 36/37, and this citation points to the wrong entry in the references section.
6. Line 308: Probably worth adding this citation to the poleward shift comment:
<https://www.nature.com/articles/s41561-021-00859-1>
7. Lines 310-311: Some awkward wording here. Might simplify to "which likely makes additional SST warming unavoidable."
8. Line 312: I would specify what precisely needs to be bolstered, i.e., high-resolution models capable of better simulating complex TC core processes.

That is it. Well done. I look forward to hopefully seeing this paper in release soon.

Response to Reviewers

Reviewer #1 (Remarks to the Author):

1. RI is a statistical concept, which is defined as approximately the 95th percentile of 24-h intensity changes. RI was suggested to be more likely just the tail of a distribution and not a distinct physical process (Kowch & Emanuel, 2015). The authors' previous study (Bhatia et al. 2019) found TC intensification rate increased during the period 1982-2009. Both negative slopes for the lower percentiles and positive slopes for the upper quantiles in the quantile regression were found in Bhatia et al. 2019 (Figure 2), which suggested a broadening of the intensity change distribution. Due to the broadening of the intensity change distribution, the 95th percentile of 24-h intensity changes increases from 30 knots to a higher rate, for instance, 35 knots. Therefore, the increased RI ratio, which was defined as the number of 24-hour intensity changes greater than 30 knots divided by all 24-hour intensity changes in the paper, suggests less storms maintaining a steady intensity and more storms exhibiting intensification rate higher than 30 knot. The possible influences of changing thermodynamic environments on less storms maintaining a steady intensity and more storm exhibiting high intensification rate should be discussed.

The reviewer brings up an excellent point that was highlighted in Figure 2 of Bhatia et al. (2019): the analyzed data suggest that more storms are experiencing high intensification rates and broadening the intensity change distribution. As a result, we have added to the manuscript lines 117-121: "The change in the proportion of intensity changes undergoing RI in these basins is part of an overall broadening of the intensity distribution and not just an increase in intensity changes greater than 30 knots. Supplementary Fig. 1 is formulated using a similar methodology to Figure 2 in Bhatia et al. (2019) and shows that the majority of the upper quantiles are increasing and lower quantiles are decreasing, reflecting less storms maintaining a steady intensity."

In addition to this new figure, we have also added lines 244-246, "Figure 1, Supplementary Figure 1, and Figure 4 align well with theory that the distribution of TC intensity and intensification should shift to higher values with more conducive thermodynamic environments (Emanuel 2000; Kossin et al. 2020)."

References

- Emanuel, K. A Statistical Analysis of Tropical Cyclone Intensity. *Monthly Weather Review* **128**, 1139-1152 (2000). [https://doi.org:10.1175/1520-0493\(2000\)128<1139:asaotc>2.0.co;2](https://doi.org:10.1175/1520-0493(2000)128<1139:asaotc>2.0.co;2)
- Kossin, J. P., Knapp, K. R., Olander, T. L. & Velden, C. S. Global increase in major tropical cyclone exceedance probability over the past four decades. *Proceedings of the National Academy of Sciences of the United States of America* **117**, 11975-11980 (2020). <https://doi.org:10.1073/pnas.1920849117>

2. The paper applied the same method as Kaplan and Demaria (2003) to identify the environmental thresholds for TC RI. RI cases in Kaplan and Demaria (2003) were relative to non-RI cases, which included weakening and neutral storms. It is sure that some environmental conditions are different between RI and weakening/neutral storms. However, Hendricks et al. (2010) found that the environment of RI TCs and intensifying TCs is quite similar. Their study suggested, given the environment is favorable, the rate of intensification is not critical dependent on the environmental conditions and RI is mostly controlled by internal dynamical processes. The authors need to discuss the discrepancy.

This comment reflects that there was a bit of confusion on the use of the thresholds and ramifications of the analysis so we will clarify here and in the text. Firstly, we did not follow the exact same method as Kaplan and DeMaria (2003). We clarified this on lines 190-193 with the bolded text, “**However, rather than taking the mean of the initial conditions for RI and non-RI cases to define the critical threshold for an environmental parameter**, we solve the logit equation to attain the critical threshold that corresponds to the probability of RI in each basin.” Therefore, our approach uses logistic regression to objectively determine what value corresponds to the threshold of RI and then we report on how RI probability differs on either side of that threshold. Note that intensifying, weakening, and rapidly intensifying cases exist on both sides of the threshold, but our analysis shows that weaker shear, stronger PI, higher RH, and higher SST are all beneficial for RI. Figure 3b further confirms this result.

Note that the Hendricks et al. (2010) study is an important study, but the methodology and analysis is quite different than what we present here. The authors look at a time period of 2003-2008 in 2 ocean basins using the NOGAPS dataset. Additionally, they use a 10 degree x 10 degree box around storms. Here, we use a dataset 6 times longer over more basins and use modern reanalyses datasets and spectral filtering to define more robust domains around storms for analysis.

Regardless, we add this text on lines 212-220 to make sure this is clear to our readers, “The documented relationship between TC intensification and the storm environment is supported by theory and numerous modelling experiments (Emanuel 1991, Done 2022, Ng and Vecchi 2020, Tao and Zhang 2015). Although internal dynamical processes of TCs are also important for RI (Hendricks et al. 2010), they are challenging to quantify and predict even with high-resolution numerical weather prediction models (Trabing and Bell 2020). Additionally, it is not currently possible to assess how the internal dynamical processes will change with global warming using global climate models while large-scale environmental conditions are well-observed and captured by global climate models. Hence, we aim to develop understanding on how the environmental conditions surrounding TCs, rather than TCs’ mesoscale processes, respond to a changing climate.”

References

- Done, J. M., Lackmann, G. M. & Prein, A. F. The response of tropical cyclone intensity to changes in environmental temperature. *Weather Clim. Dynam.* **3**, 693-711 (2022). <https://doi.org:10.5194/wcd-3-693-2022>
- Hendricks, E. A., Peng, M. S., Fu, B. & Li, T. Quantifying Environmental Control on Tropical Cyclone Intensity Change. *Monthly Weather Review* **138**, 3243-3271 (2010). <https://doi.org:10.1175/2010mwr3185.1>
- Kaplan, J. & DeMaria, M. Large-Scale Characteristics of Rapidly Intensifying Tropical Cyclones in the North Atlantic Basin. *Weather and Forecasting* **18**, 1093-1108 (2003). [https://doi.org:10.1175/1520-0434\(2003\)018<1093:lcorit>2.0.co;2](https://doi.org:10.1175/1520-0434(2003)018<1093:lcorit>2.0.co;2)
- Emanuel, K. A. The Theory of Hurricanes. *Annual Review of Fluid Mechanics* **23**, 179-196 (1991). <https://doi.org:10.1146/annurev.fl.23.010191.001143>
- Ng, C. H. J. & Vecchi, G. A. Large-scale environmental controls on the seasonal statistics of rapidly intensifying North Atlantic tropical cyclones. *Climate Dynamics* **54**, 3907-3925 (2020). <https://doi.org:10.1007/s00382-020-05207-4>
- Ramsay, H. A., Singh, M. S. & Chavas, D. R. Response of Tropical Cyclone Formation and Intensification Rates to Climate Warming in Idealized Simulations. *Journal of Advances in Modeling Earth Systems* **12**, e2020MS002086 (2020). <https://doi.org:https://doi.org/10.1029/2020MS002086>
- Tao, D. & Zhang, F. Effects of vertical wind shear on the predictability of tropical cyclones: Practical versus intrinsic limit. *Journal of Advances in Modeling Earth Systems* **7**, 1534-1553 (2015). <https://doi.org:https://doi.org/10.1002/2015MS000474>
- Trabing, B. C. & Bell, M. M. Understanding Error Distributions of Hurricane Intensity Forecasts during Rapid Intensity Changes. *Weather and Forecasting* **35**, 2219-2234 (2020). <https://doi.org:10.1175/waf-d-19-0253.1>

Reviewer #2 (Remarks to the Author):

In this work the authors analyze the occurrence of the largest tropical cyclone intensification rates in the observational datasets of the last decades and show that their increase is mainly linked to the increase in sea surface temperature and that such increase has likely been affected by anthropic activities. The study is well designed, and the analysis generally supports the conclusions. The manuscript is very well written and includes the relevant details. However, in the present form it does not add much information to what we already know about tropical cyclone and anthropogenic climate change.

The fact that SST is a fundamental driver of tropical cyclone intensification is a well-known thing that dates back to the work by Malkus and Riehl, 1960. More recently, Xu et al. 2016 and Xu and Wang 2018 showed the existence of a maximum potential intensification rate that primarily depends on SST. It is also well known that SSTs have been increasing over the last decades and that anthropic activities play role in this increase. The novelty in this manuscript is that not only TC intensification rate but also rapid intensification events are increasing, primarily driven by the increasing SSTs. The scientific literature has already highlighted the existence of the upward trend in extreme rapid intensification events over the last 30 years (Klotzbach et al. 2022).

We would like to share some clarifying information with this reviewer to help put this study in context. This study leverages new datasets, climate model output, and analysis techniques to produce 3 novel conclusions. First, higher intensification rates for global tropical cyclones are becoming more probable between 1982-2017 which is largely driven by anthropogenic forcing. Second, tropical cyclones are experiencing more favorable thermodynamic environments during this period which suggests this uptick in rapid intensification likelihood is physically driven. Finally, the improvement of tropical thermodynamic environments for rapid intensification of tropical cyclones are likely caused by anthropogenic forcing agents.

Furthermore, the reviewer is overlooking a crucial novel aspect of our work. Our study, in contrast to other works, such as Klotzbach et al. (2022), explicitly examines the question of whether the observed changes are larger than expected from natural variability alone (using climate models to estimate the potential natural variability influence). We also, in contrast to other studies, use climate models to attempt to simulate the response of RI to changes in anthropogenic forcing, to assist in establishing causal mechanisms for the RI changes (going beyond just SST). This approach of using climate models to attempt to establish causal mechanisms (climate forcings or natural variability) as opposed to showing simple correlations with SSTs (where we do not know what caused the SST changes), set our study well apart from previous works. In that regard, we also note that Klotzbach et al. provides no attribution conclusions for the observed changes in terms of the whether and how anthropogenic forcing was responsible for the RI changes. To make these significant advancements clearer in the text, we add on lines 320-322, "This study leverages new datasets, climate model output, and analysis techniques to explicitly examine the question of whether climate change has contributed to the observed changes in TC intensification and environments surrounding TCs." In our introduction, we also have lines 70-76 which address these topics.

In order to put the results presented in this manuscript in perspective, we need to consider the definition of rapid intensification events that is used, i.e., the number of events with 24hr intensity change larger than a fixed threshold. Considering that intensification rates have increased, mainly driven by higher SSTs and, related to SSTs, higher potential intensity, it is expected that all the intensification rates, including the higher ones, shifted to larger values. The authors are well aware of this expectation,

and in fact they state that what is missing in the literature is an analysis on whether changes in storm tracks and variability on weather time scales could prevent TCs from experiencing large scale environmental conditions, such as the increased SST, that are more conducive for RI (lines 165-167). Their conclusion is that this did not happen. However, they did not fully investigate the issue: It would be interesting to know whether the distribution of intensification rate changed shape or not, whether the high end tail got higher or lower. In short, are data consistent with a shift of the intensification rate pdf to larger values or not? I encourage the authors to assess this.

The reviewer brings up an excellent point that was highlighted in Figure 2 of Bhatia et al. (2019): the analyzed data suggest that more storms are experiencing high intensification rates and broadening the intensity change distribution. As a result, we have added to the manuscript lines 117-121: “The change in the proportion of intensity changes undergoing RI in these basins is part of an overall broadening of the intensity distribution and not just an increase in intensity changes greater than 30 knots. Supplementary Fig. 1 is formulated using a similar methodology to Figure 2 in Bhatia et al. (2019) and shows that majority of the upper intensification quantiles are increasing, reflecting less storms maintaining a steady intensity.”

In addition to this new figure, we have also added lines 244-246, “Figure 1, Supplemental Figure 1, and Figure 4 align well with theory that the distribution of TC intensity and intensification should shift to higher values with more conducive thermodynamic environments (Emanuel 2000; Kossin et al. 2020).”

References

Emanuel, K. A Statistical Analysis of Tropical Cyclone Intensity. *Monthly Weather Review* **128**, 1139-1152 (2000). [https://doi.org:10.1175/1520-0493\(2000\)128<1139:asaotc>2.0.co;2](https://doi.org:10.1175/1520-0493(2000)128<1139:asaotc>2.0.co;2)

Kossin, J. P., Knapp, K. R., Olander, T. L. & Velden, C. S. Global increase in major tropical cyclone exceedance probability over the past four decades. *Proceedings of the National Academy of Sciences of the United States of America* **117**, 11975-11980 (2020). <https://doi.org:10.1073/pnas.1920849117>

It is acknowledged the fact that forecast errors for Rapid Intensification (RI) events are 2-3 times larger than for non-RI events, which indicates that some factors that favour their occurrence have not been identified yet, but this paper does not shed light onto this. Actually, the fact that the forecast error for RI events is large indicates that the large scale conditions that are known to favour their occurrence (and that are analysed in this manuscript) do not capture the key elements that lead to RI events. This should be highlighted.

Yes, we agree the forecast error for RI events is larger than non-RI events. However, several studies show there is still skill in predicting in RI and environmental variables explain some of the variance. Most recently, see Cangialosi et al. (2020) discussion of RII.

Nevertheless, in response to the reviewer's comments, we add this text on lines 212-220 to make sure this is clear to our readers, "The documented relationship between TC intensification and the storm environment is supported by theory and numerous modelling experiments (Emanuel 1991, Done 2022, Ng and Vecchi 2020, Tao and Zhang 2015). Although internal dynamical processes of TCs are also important for RI (Hendricks et al. 2010), they are challenging to quantify and predict even with high-resolution numerical weather prediction models (Trabing and Bell 2020). Additionally, it is not currently possible to assess how the internal dynamical processes will change with global warming using global climate models while large-scale environmental conditions are well-observed and captured by global climate models. Hence, we aim to develop understanding on how the environmental conditions surrounding TCs, rather than TCs' mesoscale processes, respond to a changing climate."

References

- Cangialosi, J. P., Blake, E., DeMaria, M., Penny, A., Latta, A., Rappaport, E., & Tallapragada, V. (2020). Recent Progress in Tropical Cyclone Intensity Forecasting at the National Hurricane Center, *Weather and Forecasting*, 35(5), 1913-1922.
- Done, J. M., Lackmann, G. M. & Prein, A. F. The response of tropical cyclone intensity to changes in environmental temperature. *Weather Clim. Dynam.* **3**, 693-711 (2022). <https://doi.org:10.5194/wcd-3-693-2022>
- Hendricks, E. A., Peng, M. S., Fu, B. & Li, T. Quantifying Environmental Control on Tropical Cyclone Intensity Change. *Monthly Weather Review* **138**, 3243-3271 (2010). <https://doi.org:10.1175/2010mwr3185.1>
- Kaplan, J. & DeMaria, M. Large-Scale Characteristics of Rapidly Intensifying Tropical Cyclones in the North Atlantic Basin. *Weather and Forecasting* **18**, 1093-1108 (2003). [https://doi.org:10.1175/1520-0434\(2003\)018<1093:lcorit>2.0.co;2](https://doi.org:10.1175/1520-0434(2003)018<1093:lcorit>2.0.co;2)
- Emanuel, K. A. The Theory of Hurricanes. *Annual Review of Fluid Mechanics* **23**, 179-196 (1991). <https://doi.org:10.1146/annurev.fl.23.010191.001143>
- Ng, C. H. J. & Vecchi, G. A. Large-scale environmental controls on the seasonal statistics of rapidly intensifying North Atlantic tropical cyclones. *Climate Dynamics* **54**, 3907-3925 (2020). <https://doi.org:10.1007/s00382-020-05207-4>

Ramsay, H. A., Singh, M. S. & Chavas, D. R. Response of Tropical Cyclone Formation and Intensification Rates to Climate Warming in Idealized Simulations. *Journal of Advances in Modeling Earth Systems* **12**, e2020MS002086 (2020). <https://doi.org/10.1029/2020MS002086>

Tao, D. & Zhang, F. Effects of vertical wind shear on the predictability of tropical cyclones: Practical versus intrinsic limit. *Journal of Advances in Modeling Earth Systems* **7**, 1534-1553 (2015). <https://doi.org/10.1002/2015MS000474>

Trabing, B. C. & Bell, M. M. Understanding Error Distributions of Hurricane Intensity Forecasts during Rapid Intensity Changes. *Weather and Forecasting* **35**, 2219-2234 (2020). <https://doi.org/10.1175/waf-d-19-0253.1>

Minor comments:

Rapid Intensification is identified as the 24hr intensity change above the 95th percentile on line 66 and above 30 knots on line 109.

Yes, we define the metric RI ratio, and we cite the literature definition. The 30-knot threshold is used in our analysis to promote consistency spatially and temporally with our approach. The 95th percentile varies based on basin and time period which is not ideal for trend analysis.

Fig.1: The shading in the fig comes from the perturbations added to the intensities before computing intensification rates, but I don't see how is the statistical significance of the slope evaluated. In other words, is the null hypothesis of no trend rejected? Please add some description of those details.

On line 111-113, we modified the text to address this suggestion, "Throughout all basins, there are significant (rejecting the null hypothesis of no trend at the $p=0.05$ significance level) upward trends in RI ratio defined using IBTrACS data (Fig. 1), which agrees well with recent studies^{6,10}."

Selection of tracks: The analysis is restricted to TC with track starting and ending position below 40° of latitude (line 396). The authors conclude that track variability (such as a shift toward more poleward locations) is unlikely to prevent additional increases in TC intensification with further anthropogenic warming (lines 307-309). Such conclusion however might not be supported by the present analysis, as the authors are not considering TC that end at latitudes higher than a fixed threshold. In theory, TCs with tracks extending at higher latitudes could actually experience a decrease in the occurrence of rapid intensification events. Although I don't expect the results to significantly change, the inclusion of such storm seems to me necessary, in light of the goal of the study.

The goal of this manuscript is to analyze tropical cyclones and their intensification pathways. These higher latitude storms often have extratropical characteristics (and are governed by different dynamical mechanisms). Thus, we added this text to lines 423-425. "We do not examine TC intensity changes above 40° latitude because storms typically complete extratropical transition (Bieli et al. 2019) and achieve their lifetime maximum intensity equatorward of this latitude (Kossin et al. 2014)." Additionally, in Bhatia et al. (2019), we checked the sensitivity of results at different latitude thresholds and found negligible impact.

References

Bieli, M., Camargo, S. J., Sobel, A. H., Evans, J. L. & Hall, T. A Global Climatology of Extratropical Transition. Part I: Characteristics across Basins. *Journal of Climate* **32**, 3557-3582 (2019). <https://doi.org:10.1175/jcli-d-17-0518.1>

Kossin, J. P., Emanuel, K. A. & Vecchi, G. A. The poleward migration of the location of tropical cyclone maximum intensity. *Nature* **509**, 349-352 (2014). <https://doi.org:10.1038/nature13278>

Reviewer #3 (Remarks to the Author):

Dear Authors:

General comments

1. Lines 161-179: I would like to see more discussion of what PI is both conceptually and physically when it is introduced, especially as SST and PI are both used as environmental proxies for the remainder of the paper, and of course there is a dependency in play there. I don't take any issues with the methodology of the study, but I do think some additional context in the introduction of the concepts would be beneficial in order to make the link between anthropogenic influence and RI outcomes clearer.

Thank you for this recommendation, we have added this text on lines 173-180: "PI and SST are related metrics, but PI is uniquely impacted by the tropospheric profile of temperature and moisture. Calculated using the large-scale environment, PI represents an upper limit of TC intensity (Emanuel 1988) that is derived from the thermodynamic disequilibrium between the surface of the ocean and the upper atmosphere (Vecchi and Soden 2007). When PI increases, the theoretical intensity range for a storm expands and greater 24-hour intensity changes are possible. For these reasons and because of the ongoing research on the evolution of the relationship of PI and SST under climate change, we include both environmental variables in our analysis analysis (Ramsay and Sobel 2011).

References

Emanuel, K. A. The Maximum Intensity of Hurricanes. *Journal of Atmospheric Sciences* 45, 1143-1155 (1988). [https://doi.org/10.1175/1520-0469\(1988\)045<1143:tmioh>2.0.co;2](https://doi.org/10.1175/1520-0469(1988)045<1143:tmioh>2.0.co;2)

Vecchi, G. A. & Soden, B. J. Effect of remote sea surface temperature change on tropical cyclone potential intensity. *Nature* 450, 1066-1070 (2007). <https://doi.org/10.1038/nature06423>

Ramsay, H. A. & Sobel, A. H. Effects of Relative and Absolute Sea Surface Temperature on Tropical Cyclone Potential Intensity Using a Single-Column Model. *Journal of Climate* 24, 183-193 (2011). <https://doi.org/10.1175/2010jcli3690.1>

2. Lines 275-297: I think more needs to be said and shown here beyond "further research is required" to address the major differences between PI trends in the climate models and reanalyses. The physical explanation offered in the second paragraph seems highly credible, but I believe a supplemental figure comparing average tropical temperatures with height between the reanalyses and CMIP6 would go a long way for the reader in making it convincing. As is, that paragraph seems needlessly speculative.

Thank you for this suggestion. We have created a supplementary figure to address this suggestion. We added on lines 313-318, "Supplementary figure 4 displays trends in the tropical vertical temperature profiles in CMIP6, MERRA-2, and ERA5 between 1982-2014. The discrepancy between the trends in the upper troposphere appears to support Keil et al. and partially explain the differing trends in CMIP6 and the reanalysis data. This finding is particularly meaningful and demands further research because it implies that future increases of TC intensities and RI could be underestimated by current climate model-based projections that contain this bias."

Specific comments

1. Line 46: I would clarify that "worst forecasts" refers to high intensity errors at relatively short operational lead times.

We made a change to clarify but for simplicity and word limits, we did not include anything about the lead time. "Tropical cyclone rapid intensification events often cause destructive hurricane landfalls because they are associated with the strongest storms and forecasts with the highest errors."

2. Line 71-72: Add "intensity" before "forecast errors."
Corrected.

3. Lines 103/125: These lines repeat each other.
Removed the second instance.

4. Line 158: Might be worth citing Walsh et al 2013 here?
Added.

5. Line 289: Something went wrong with citations 36/37, and this citation points to the wrong entry in the references section.

Corrected.

6. Line 308: Probably worth adding this citation to the poleward shift comment: <https://www.nature.com/articles/s41561-021-00859-1>

Added.

7. Lines 310-311: Some awkward wording here. Might simplify to "which likely makes additional SST warming unavoidable."

Corrected.

8. Line 312: I would specify what precisely needs to be bolstered, i.e., high-resolution models capable of better simulating complex TC core processes.

Added additional text on lines 335-341, "Given the observed increase in RI ratio and the possibility for a continuing trend into the future, bolstering the development of high- resolution models capable of better resolving mesoscale processes and the environments surrounding TCs should be a societal priority. Additionally, our conclusions suggest climate change has already contributed to the larger proportion of rapidly intensifying hurricanes and highlights the immediate need to build more coastal resilience to prepare against these dangerous events."

REVIEWERS' COMMENTS

Reviewer #1 (Remarks to the Author):

The study examined the question of whether climate change has contributed to TC rapid intensification and environments surrounding TCs. The revised manuscript has answered the questions and concerns raised by reviewers and is ready to be published.

Reviewer #2 (Remarks to the Author):

Thanks, the authors have satisfactorily answered to all my comments. Fig. S1 was especially useful. I think it leaves wide open questions, as the fact that there is a change in the intensification rate distribution (it's getting wider, with fatter tails, and less storms keep a nearly stationary intensity) is not consistent with the general increase in occurrence of large scale conditions that favour rapid intensification. The large scale conditions are changing to favour stronger intensification rates (consistent with more RI events), but actually a larger portion of storms is experiencing weakening. How can this be reconciled? I think that this "paradox" should be mentioned somewhere in the manuscript. But I leave to the authors the choice of whether to include a sentence on this or not.

Reviewer #3 (Remarks to the Author):

The authors have satisfied my comments and questions from the first round of review. As previously, I support the publication of this useful and interesting manuscript.

Response to Referees

Reviewer #1 (Remarks to the Author):

The study examined the question of whether climate change has contributed to TC rapid intensification and environments surrounding TCs. The revised manuscript has answered the questions and concerns raised by reviewers and is ready to be published.

Thank you for the positive feedback and for the original helpful edits.

Reviewer #2 (Remarks to the Author):

Thanks, the authors have satisfactorily answered to all my comments. Fig. S1 was especially useful. I think it leaves wide open questions, as the fact that there is a change in the intensification rate distribution (it's getting wider, with fatter tails, and less storms keep a nearly stationary intensity) is not consistent with the general increase in occurrence of large scale conditions that favour rapid intensification. The large scale conditions are changing to favour stronger intensification rates (consistent with more RI events), but actually a larger portion of storms is experiencing weakening. How can this be reconciled? I think that this "paradox" should be mentioned somewhere in the manuscript. But I leave to the authors the choice of whether to include a sentence on this or not.

Thank you for the positive feedback and for the original helpful edits. We agree this is an interesting topic of further research and we will leave it for future work. We do not necessarily see the rapid weakening as a paradox as the higher lifetime maximum intensities of storms post-RI provide more opportunities for rapid weakening as a storm moves to an unfavorable environment.

Reviewer #3 (Remarks to the Author):

The authors have satisfied my comments and questions from the first round of review. As previously, I support the publication of this useful and interesting manuscript.

Thank you for the positive feedback and for the original helpful edits.